# Posterior Fossa Approaches Using the Leksell Vantage Frame with a Virtual Planning Approach in a Series of 10 Patients—Feasibility, Accuracy, and Pitfalls

**DOI:** 10.3390/brainsci12121608

**Published:** 2022-11-24

**Authors:** Marie T. Krüger, Alexis P. R. Terrapon, Alexander Hoyningen, Chan-Hi Olaf Kim, Arno Lauber, Oliver Bozinov

**Affiliations:** 1Department of Neurosurgery, The National Hospital for Neurology and Neurosurgery, Queen Square, London WC1N 3BG, UK; 2Department of Neurosurgery, Cantonal Hospital, 9007 St. Gallen, Switzerland; 3Department of Stereotactic and Functional Neurosurgery, University Medical Centre, 79106 Freiburg, Germany; 4Department of Radiology and Nuclear Medicine, Cantonal Hospital, 9007 St. Gallen, Switzerland

**Keywords:** stereotactic biopsy, frame-based, Leksell Vantage, posterior fossa, planning software

## Abstract

The open-face design of the Leksell Vantage frame provides many advantages. However, its more rigid, contoured design offers less flexibility than other frames. This is especially true for posterior fossa approaches. This study explores whether these limitations can be overcome by tailored frame placement using a virtual planning approach. The posterior fossa was accessed in ten patients using the Leksell Vantage frame. Frame placement was planned with the Brainlab Elements software, including a phantom-based (virtual) pre-operative planning approach. A biopsy was performed in all patients; in four, additional laser ablation surgery was performed. The accuracy of virtual frame placement was compared to actual frame placement. The posterior approach was feasible in all patients. In one case, the trajectory had to be adjusted; in another, the trajectory was switched from a right- to a left-sided approach. Both cases showed large deviations from the initially planned frame placement. A histopathological diagnosis was achieved in all patients. The new Leksell Vantage frame can be used to safely target the posterior fossa with a high diagnostic success rate and accuracy. Frame placement needs to be well-planned and executed. This can be facilitated using specific software solutions as demonstrated.

## 1. Introduction

Posterior cranial stereotactic approaches often offer a shorter and safer way to access lesions in the cerebellum, cerebellar and cerebral peduncles, and other parts of the brainstem. However, frame-based procedures can be challenging due to restricted entry and deep target points. Nevertheless, multiple studies show that this approach can be performed with high accuracy and few complications [1,2,3,4,5,6] using popular, commercially available systems, such as the Leksell G, Riechert Mundinger, Brown-Roberts-Wells (BRW) and Cosman-Roberts-Wells (CRW) frames [7,8,9,10,11,12]. These frames allow for adjustments that make a posterior fossa approach more feasible [7,13]. For example, for the Leksell G-frame, Capitanio et al. and Horisawa et al. proposed attaching the Leksell G-frame upside down to increase the access window [13,14]. Neal et al. proposed the use of longer posts to increase the size of the biopsy window [15]. Spiegelman et al. and Guthrie and colleagues suggested a 90° or 180° rotation respectively, before attachment of the CRW or CBW frame [10,11].

With its rigid design, the Leksell Vantage frame has increased stability, at the cost of compromising flexibility that does not permit the above-described modifications. The arc-shaped posterior aspect of the frame, limits the “access window” to 14 × 7 cm (Figure 1A). Moreover, the fixation attachment (Figure 1C) poses a new limitation to the posterior ring angle, since it can block the ring range of the arc system (Figure 1B,B’). This can be overcome, to some degree, by rotating the stereotactic arc through 180° (to a left-lateral origin approach), increasing the ring range by 10–20°. However, ultimately, this is also blocked by the attachment ring or holder (Figure 1D,D’). 

These limitations can be overcome by tailored frame placement that is specific to the intended target and trajectory. The access window must be placed such that the trajectory is clear of the frame outline and allows a feasible ring angle. This requires planning how the frame should be attached to the head. 

The Brainlab Elements (Munich, Germany) planning software allows three-dimensional visualization and virtual manipulation of patient-specific images and segmented objects. A segmented CT image of a Vantage frame can be adjusted virtually in relation to patient-specific images, until a viable trajectory can be found that utilizes the available access window at the permissible ring and arc angles, thus establishing well-defined entry and target points. This virtual plan can then be used as a model for actual frame placement.

In this study, we present a series of ten consecutive patients in whom a posterior fossa approach was performed using the Leksell Vantage frame. The steps required to perform a virtual approach to frame placement and trajectory planning are demonstrated. The accuracy, feasibility, and safety of replicating this in the operating theatre are examined, and potential pitfalls/limitations of this concept are discussed. Our virtual planning method enables to use of the new Leksell Vantage frame to target posterior fossa lesions safely and effectively.

## 2. Materials and Methods

### 2.1. Basic Considerations

Since the rigid design of the Vantage frame cannot be changed, the required entry point and trajectory must become accessible and feasible through appropriate frame placement. 

There are three levels of freedom for frame placement: pitch, yaw, and roll (Figure 2A). To avoid collision with the lower and upper boundaries the pitch (sagittal angle) can be adjusted (Figure 2B). To avoid collision with the vertical side posts, the yaw (axial rotation) can be adjusted (Figure 2C,D). For most target coordinates the ring should be between 160–80° for right-lateral origin or between 340–6° for left-lateral origin approaches (see Table 1). In our experience, the roll should be kept to a minimum. 

### 2.2. Determination of Available Ring and Arc Angles

To determine programmable ring and angle coordinates, we performed a phantom study using a Leksell Vantage frame. Most applied X, Y and Z coordinates were programmed in 10 mm steps (Table 1). We then recorded feasible Ring and Arc angles for all combinations when (a) the guide holder is brought past the frame completely (Table 1, upper row; Appendix A) and (b) only the drill bit is brought past the frame (Table 1, italic numbers in a lower row; Appendix A). The most conservative angles within the 10 mm range were chosen. The results are displayed in Table 1.

### 2.3. Patients

All patients who underwent stereotactic biopsy of posterior fossa lesions using the Leksell Vantage (Elekta, Stockholm) frame from December 2020 to June 2022, were included. The indication was approved by an interdisciplinary team of neurosurgeons, neurologists, oncologists, and neuroradiologists. A posterior approach was favored over a frontal approach due to the location of the lesion and/or for safety reasons. Demographic and histological data were collected from medical records.

### 2.4. Virtual Frame-Placement

Planning was performed using the Brainlab Elements^®^ software (Brainlab, Munich) as follows:1.One-time preparation:The Vantage frame was fixed to a skull phantom and its CT fiducial box attached;A CT scan (0.7 mm slice thickness, Somatom Force, Siemens Healthineers, Munich, Germany) was obtained of the above ensemble (P-CT);The *object segmentation* tool was used to segment the frame and its fiducial box.2.Patient-specific planning:The *trajectory planning* tool was used to plan a trajectory to the surgical target on preoperatively acquired, patient-specific MR images;The *fusion* tool was used to co-register the P-CT with its segmented frame to patient-specific images (Figure 3A,B);Coordinates were checked to see whether frame alignment adjustment was required (Figure 3C,D, Table 1);The *fusion* tool was used, when necessary, to adjust frame height (in coronal view), yaw (in axial view) and pitch (in sagittal view) in relation to the patient-specific images (Figure 3E,F);Coordinates were re-checked (Figure 3G,H) and the process repeated until all coordinates and trajectory were feasible;The *trajectory planning* tool was used to display the spatial relationship between the frame and the patient-specific 3D surface render. Frontal, posterior, and lateral views were saved and printed (Figure 3E`,E``,F`);The location of the planned target and entry point as well as anterior and posterior posts from skin landmarks was measured and recorded.

### 2.5. Actual Frame-Placement

Frame placement was performed under local (*n* = 3) or general anesthetic (*n* = 7) in a semi-sitting position. The printed virtual planning images were used to mark the following: (a) entry point; (b) lateral surface projection of the target point; (c) a line was drawn between the two (Figure 4A); the pin site of the (d) posterior and (e) anterior pole were marked (Figure 4A–D). The anterior post was positioned such that the angle of the frame was steeper than the projected trajectory line angle (Figure 4B, dotted red and blue line angle) and the entry point was within the feasible window (Figure 4B,D, green bar). The frame was then fixed using the Elekta pins as per the standard of care. Figure 5 shows the attached frames on all patients in lateral view. Figure 3 shows an example of patient no. 3. Note that for left-sided approaches the angle can be >180°.

### 2.6. Actual Surgical Planning

After fixation of the frame, all patients underwent a CT scan (0.7 mm slice thickness, Somatom Force, Siemens Healthineers, Munich, Germany) with the CT fiducial module attached (Elekta, Sweden). This CT was registered to the preoperative planning MRI (T1-weighted MPRAGE with contrast agent (0.9 mm thickness; 3 Tesla, Magnetom Skyra fit, Siemens Healthineers), and coordinates were calculated. The feasibility of ring and arc coordinates was checked using Table 1 and by using an extra Vantage frame as a phantom. If these were not feasible, adjustments were made to the trajectory until a feasible one was obtained. 

### 2.7. Surgical Procedure

Patients were placed in a sitting/half-sitting position (Appendix A). The frame was attached to the Mayfield holder. After skin preparation and draping, a 5 mm skin incision and a 3.2 mm twist drill were placed in line with the planned trajectory. Three to five biopsies were taken from the target using a 2.1 mm side-cutting biopsy needle (Elekta, Sweden). Wounds were closed using staples or sutures. In case of an additional LITT procedure, a laser catheter (Visualase, Medtronic) was placed after removal of the biopsy needle, and patients were transferred to the MRI suite for real-time visualization of tumor ablation. 

### 2.8. Evaluation of Accuracy between Virtual and Actual Frame Placement

In all cases, the target coordinates as well as ring and arc settings of the virtual frame planning were compared to those calculated after actual frame placement; the differences were calculated and recorded. The need for intraoperative adjustments to frame placement or surgical trajectory and any adverse events were also recorded.

### 2.9. Ethics

This study was approved by the local Ethics committee (EKOS 22/139). All patients gave informed consent to be included in the study.

## 3. Results

### 3.1. Patients

Ten patients, 6 male, mean age 32 (range: 2–83) years, were included. The average lesion volume was 5.4 (range: 0.17 to 14.4) cm^3^. Five lesions were in the cerebellum/cerebellar peduncle (Appendix A), three primarily in the pons, and two primarily in the medulla (Appendix A). In all cases a biopsy was performed, and four patients received additional LITT treatment (patients 3, 5, 8, and 9). 

### 3.2. Histological Results

A histopathological diagnosis was achieved in all patients. All results are listed in Table 2.

### 3.3. Accuracy of Frame-Placement

The mean (range) difference between the planned and the actual trajectory values were as follows: ring: 3.9° (range 1–12°); arc: 5.95° (0.5–12.5°); target location: X coordinate: 3.3 (0.5–6.5) mm; Y coordinate: 6.1 (0.5–14.5) mm; Z coordinate: 11.6 (4–23.5) mm.

In all cases, surgery could be performed without the need to re-apply the frame. In 8/10 cases no adjustment of the surgical trajectory was necessary (Table 3). In patient 7, the frame was inaccurately placed in the Y and Z axes, both in an unfavorable direction, rendering the actual ring value too high. To adjust for this, the entry was altered slightly to decrease the ring by 2°. Patient 10 showed the largest deviation from the virtually planned frame placement with a deviation of 1.5 cm in Y and 2.3 cm in Z. The Z coordinate was not programmable and the target within the lesion had to be moved, resulting in a ring angle > 180°. Instead of re-applying the frame, it was possible to change to a left-origin lateral approach. This allowed for a larger ring angle, making the trajectory feasible. 

### 3.4. Surgical Complications

There was no mortality and no new permanent neurological deficit from the biopsy procedures. In the two-year-old patient (pat. no. 6), an asymptomatic fracture developed at one of the pin sites (Appendix A). One intracerebral haemorrhage (pat. no. 8) caused transient ataxia that completely resolved within four weeks (Appendix A–E). In this patient, the radiological diagnosis of urothelial metastasis later was overturned by a histological diagnosis of hemangioblastoma. There were no other complications or adverse events. 

## 4. Discussion

The rigid design of the Vantage frame reduces versatility when compared to other stereotactic frames and imposes some additional limitations due to the frame holder and fixation attachment, especially when considering posterior approaches. This case series demonstrates how these limitations can be overcome through virtual planning of frame placement. 

### 4.1. Feasibility and Accuracy of Frame Placement

To evaluate the feasibility of the virtual planning approach, we compared virtual coordinates with actual coordinates before any adjustments were made. These differences do not reflect the accuracy of the procedure itself and are purely an estimate of how closely actual frame placement can come to virtual frame placement and how much inaccuracy is tolerated without having to re-apply the frame. 

In 8/10 patients the frame was placed with an inaccuracy of less than one cm for most target points. In all these patients no further adjustments were necessary to perform the surgery. In the other 2 patients the initially received coordinates were not feasible but adjustments to the planned trajectory, target, or approach allowed the procedure to be performed without the need for re-applying the frame. In the second patient, the frame was placed more than 2 cm higher than planned so that the z value was not programmable, and the target point needed to be changed. This led to a ring angle of 183° which was not programmable. Ultimately, the entry point was adjusted, and we switched to a left-sided approach, which allowed for a larger ring angle. This allowed us to proceed without the need to re-apply the frame. This example shows how slight misplacements of the frame may require adjustment and how important it is to pre-plan frame placement virtually to avoid the need to re-apply the frame. This would include the need for another stereotactic CT, thus additional X-ray exposure and prolongation of the whole procedure. 

### 4.2. Frame Placement–Essentials

Table 1 is a useful resource when planning frame placement as it allows the selection of ring and arc values that are well within the most limiting angles. For example, if the X, Y, and Z coordinates allow a range of the ring of 158–178° and an arc angle of 52 to 102°, it is advisable to plan frame placement, such that the ring is around 168° and the arc around 77° to increase the chance that coordinates will still be feasible should frame placement differ strongly from the virtual plan.

#### 4.2.1. Increasing the Ring Angle

The ring angle is mainly dependent on the Y coordinate (smaller values being easier to reach) and the Z coordinate (smaller values being easier to reach) for right-sided approaches. The ring angle may be blocked by the fixation attachment. However, it is possible to remove this and to perform the procedure in a prone position, stabilizing the system using a cushion. This may allow for a few more degrees of ring angle range. However, at some point, the ring angle will be limited by the frame holder itself, which cannot be removed. 

Another way to increase the ring angle range is by applying the arc with a left-lateral origin setup (Figure 1D). This offers an additional 10–20° of ring freedom. However, for very medial entry points, the slide on the arc will be blocked by the frame holder and access to the screws on the slide will be limited (Figure 1D`). While this can be tolerated and worked around when performing a biopsy, full access to the screws without moving the arc is required when performing a LITT procedure.

#### 4.2.2. Limitations to Frame Placement

Theoretically, the frame can be attached in a 180° rotation as proposed by Guthrie et al. for the BRW frame [10]. However, in this case, the CT scan would need to be performed with the patient in the prone position. We have not yet tested this approach and it may present unforeseen logistical and software challenges, including the co-registration algorithm of the planning software. 

We have noted that, due to the design of the guide holder and the stop holder, a left-lateral arc placement is superior for trajectories starting from the left, and a right-lateral arc placement better suits those starting from the right. This is an important consideration when planning for centrally located lesions when a trajectory could be planned from either side. 

### 4.3. Adverse Events

We observed one adverse event that was directly related to the frame design and the need to place the frame with significant axial rotation. In patient 6, a 2-year-old girl, this led to a fracture of the thin temporal bone that was, thankfully, not symptomatic. However, it is not clear whether other frame systems would have avoided this problem, since rotation was required to avoid the required trajectory conflicting with the posterior posts that are also part of other frame systems [12]. Nevertheless, in retrospect, this complication may have been avoided by using the more medial screw hole available on the Vantage system. 

The other adverse event was unrelated to frame placement. The imaging and history of patient 8 were suspicious for metastasis of a known bladder carcinoma leading to a plan of biopsy followed by LITT ablation of the lesion. Following biopsy and laser placement, MRI showed severe bleeding into the lesion. Histopathology revealed a highly vascular hemangioblastoma, that would not have undergone biopsy had this differential diagnosis been our major suspicion. Thankfully, the clinical impact was limited to transient ataxia, which completely resolved after a few weeks. 

### 4.4. Limitations

The limitations of this study lie in its retrospective design and the relatively low number of patients. This does not allow for comprehensive validation of the method, but rather demonstrates its feasibility as well as the necessity to pre-plan and tailor Vantage frame placement to each target and trajectory. Table 1 shows the most conservative degrees for the most common X, Y, and Z coordinates. This means that some coordinates within the X, Y, and Z combinations may well be programmable beyond the here shown numbers. It can therefore only be used as a guide for trajectory planning but should not be trusted blindly. We recommend testing the coordinates using an extra frame and getting some experience with a phantom before performing this method on patients for the first time. 

We did not use any other commercially available planning software and it is very well possible that other software can be used in a similar way. 

## 5. Conclusions

The new Leksell Vantage frame can be used to safely target posterior fossa lesions with a high diagnostic success rate, accuracy, and low morbidity or mortality. However, due to its rigid design, the placement of the frame needs to be well-planned and executed. The virtual planning method described here appears to be a pragmatic and successful way to maximize the chances of a positive outcome. 

## Figures and Tables

**Figure 1 brainsci-12-01608-f001:**
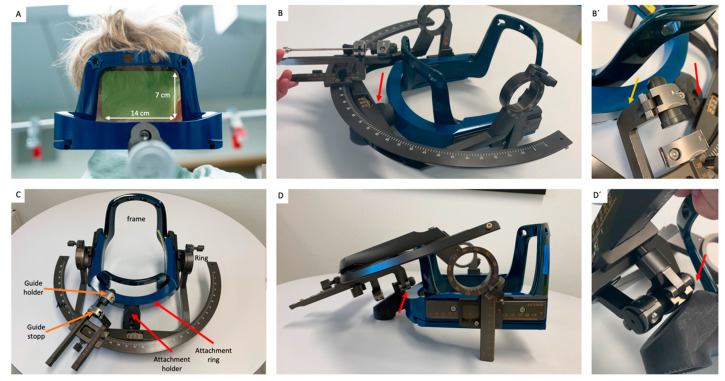
(**A**) 12 × 7 cm window for posterior fossa approaches is framed by the side posts, transverse post, and frame holder; (**B**,**B’**) right-lateral origin approach with major limitation in ring range due to frame holder (yellow arrow) or fixation attachment (red arrow); (**C**) overview of the frame system; (**D**) left-lateral origin approach (with rotation of arc through 180° on frame) increasing ring range by 10–20°; (**D’**) the fixation attachment blocks the arc system (red arrow).

**Figure 2 brainsci-12-01608-f002:**
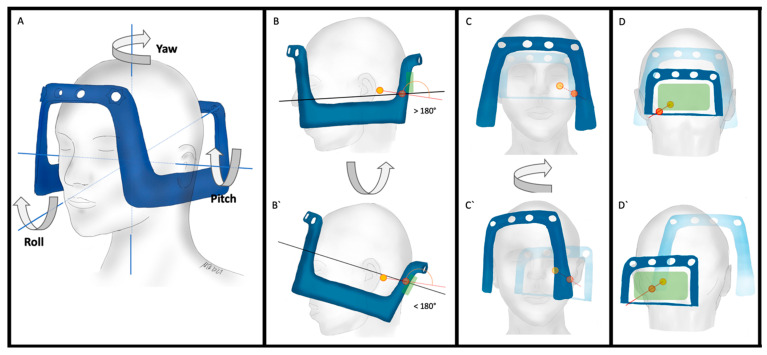
(**A**) Illustration of the three degrees of freedom: pitch (sagittal rotation), yaw (axial rotation and roll (coronal rotation); (**B**) standard frame placement with frame parallel to Reid’s baseline and limited posteroinferior trajectory with a ring angle > 180°; (**B`**) improved access to posteroinferior trajectories by changing the pitch and improving the ring angle below 180°; (**C**,**D**) frontal and posterior view of the frame with standard frame placement; (**C`**,**D`**) after adjustment of yaw to accommodate left posteroinferior entry point (red circle) to reach the target (yellow circle).

**Figure 3 brainsci-12-01608-f003:**
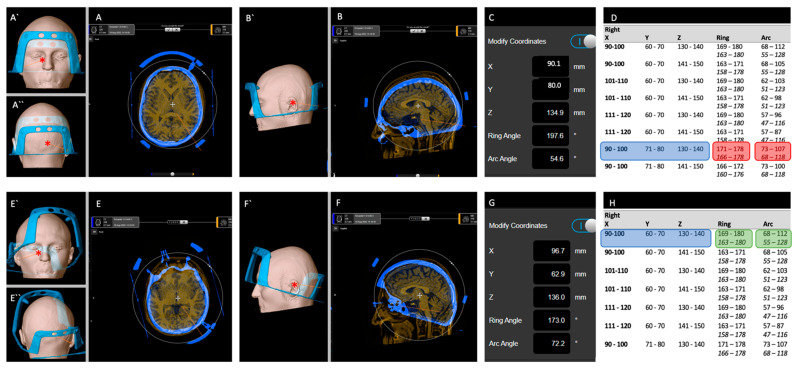
Virtual frame placement. (**A** )fusion tool view after initial co-registration with “standard” frame placement in axial and (**B**) sagittal view; (**A`**) anterior and (**A``**) posterior view; (**B`**) lateral view in “trajectory planning tool”; (**C**) coordinates with frame in this position indicate that, according to the Table 1 (**D**), ring and arc coordinates are not feasible; (**E**) After adjustment of yaw and (**F**) pitch; (**E`**) anterior and (**E``**) posterior view; (**F`**) lateral view in “trajectory planning tool”; (**G**) Coordinates are now feasible according to the Table 1 (**H**). * target; orange line: trajectory.

**Figure 4 brainsci-12-01608-f004:**
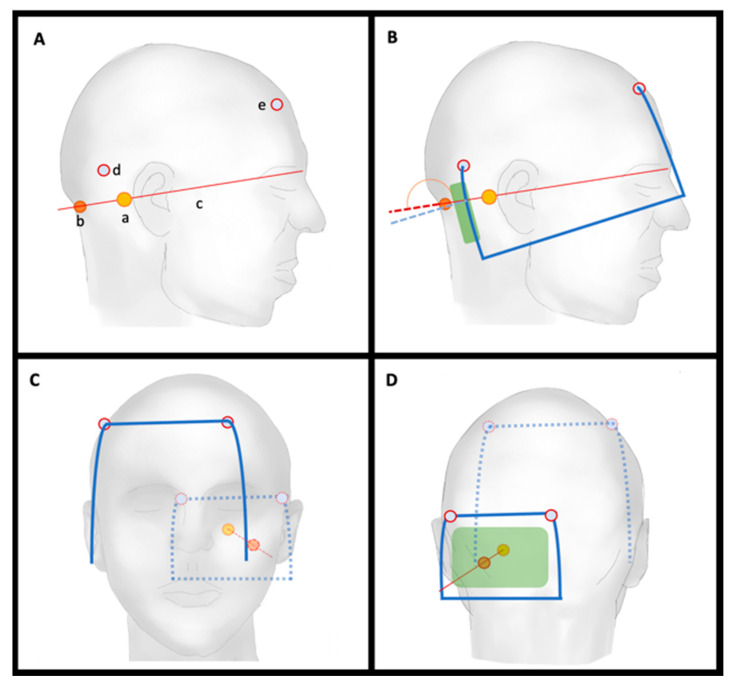
Illustration of frame-placement steps based on planned placement for right-sided approaches; (**A**) Marked (a) entry point; (b) lateral surface projection of the target point; (c) connection line between the two; (d) posterior pin site; and (e) anterior pin site; (**B**) frame attachment with angle < 180° as compared to connection line and entry point within feasible window (green area); (**C**) anterior and (**D**) posterior view.

**Figure 5 brainsci-12-01608-f005:**
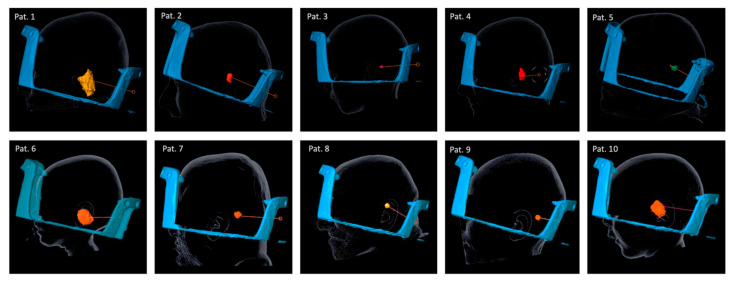
Frame placement, trajectory, and lesion in all ten patients. In patients 5, 8, and 10 a left-sided approach was performed. In all other patients, a right-sided approach was performed.

**Table 1 brainsci-12-01608-t001:** Feasible coordinates for ring and arc for the most commonly applied X, Y, and Z coordinates for right-sided and left-sided approaches. Upper row: when the guide holder is completely forwarded. Lower row (italic): when the only drill bit is completely forwarded.

Right X	Y	Z	Ring	Arc	Left X	Y	Z	Ring	Arc
**90–100**	60–70	130–140	169–180 *163–180*	68–112 *55–128*	100–110	60–75	100–115	340–353 *340–357*	69–105 *64–122*
**90–100**	60–70	141–150	163–171 *158–178*	68–105 *55–128*	100–110	60–75	116–130	344–2 *344–6*	69–105 *64–122*
**101–110**	60–70	130–140	169–180 *163–180*	62–103 *51–123*	111–120	60–75	100–115	340–353 *340–357*	77–112 *69–124*
**101–110**	60–70	141–150	163–171 *158–178*	62–98 51–123	111–120	60–75	116–130	344–2 *344–6*	77–112 *69–124*
**111–120**	60–70	130–140	169–180 *163–180*	57–96 *47–116*	121–130	60–75	100–115	340–353 *340–357*	84–117 *77–130*
**111–120**	60–70	141–150	163–171 *158–178*	57–87 *47–116*	121–130	60–75	116–130	344–2 *344–6*	84–117 *77–130*
**90–100**	71–80	130–140	171–178 *166–178*	73–107 *68–118*	100–110	76–90	100–115	341–354 *341–357*	73–104 *68–117*
**90–100**	71–80	141–150	166–172 *160–176*	73–100 *68–118*	100–110	76–90	116–130	347–359 *347–3*	73–104 *68–117*
**101–110**	71–80	130–140	171–178 *166–178*	67–99 *63–113*	111–120	76–90	100–115	341–354 *341–357*	79–108 *75–122*
**101–110**	71–80	141–150	166–172 *160–176*	67–95 *63–113*	111–120	76–90	116–130	347–359 *347–3*	79–108 *75–122*
**111–120**	71–80	130–140	171–178 *166–178*	63–92 *56–109*	121–130	76–90	100–115	341–354 *341–357*	85–119 *80–125*
**111–120**	71–80	141–150	166–172 *160–176*	63–90 *56–109*	121–130	76–90	116–130	347–359 *347–3*	85–119 *80–125*

**Table 2 brainsci-12-01608-t002:** Age, gender, tumor size, approach, location of the lesion and histopathological results.

No	Age	Sex	Size (cm^3^)	Appr.	Location	Histolopathology
1	26	F	21	Right	Pons and Medulla right	Astrocytoma Grade 3
2	45	M	1.9	Right	Mesencephalon/ Tegmentum	Lymphoma
3	7	M	0.17	Right	Cerebellum right	Ganglioglioma Grade 1
4	39	M	3.62	Right	Pons and Medulla left	Oligodendroglioma Grade 2
5	66	M	1.3	Left	Cerebellum left	Metastatic melanoma
6	2	F	14.4	Right	Medulla left	Pilocytic astrocytoma Grade 1
7	37	F	1.2	Right	Cerebellum (Vermis)	Medulloblastoma Grade 4
8	83	M	0.7	Left	Cerebellum left	Hemangioblastoma
9	10	M	0.42	Right	Cerebellum right	Pilocytic astrocytoma Grade 1
10	3	F	9.3	Left	Pons	High-grade glioma

**Table 3 brainsci-12-01608-t003:** Comparison between coordinates after “virtual” frame placement (Virt.) and after actual (Act.) frame placement. Differences (Dif.) from virtual to actual coordinates before any adjustments were made. Italic coordinated (pat. 7 and 10) shows coordinates after adjustments.

Pat	Virt. X	Y	Z	Ring	Arc	Act. X	Y	Z	Ring	Arc	Dif. X	Y	Z	Ring	Arc
1	99.5	80.5	137.5	171	68	93	74	154.5	168	78	6.5	5.5	17	3	10
2	99.5	80	133	179	66	100.5	80.5	143	177.5	71	1	0.5	10	1.5	5
3	90.5	65.5	135	172.5	73.5	92.5	71	145	177	82	2	5.5	10	5.5	2.5
4	115.5	69.5	128	165	145	116	71	140.5	153	140	0.5	1.5	13	12	5
5	127	70	108.5	347	82	121.5	66.5	120.5	350	92.5	6	3.5	12.5	3	10.5
6	102	75	142	167	110	96.5	71	151.2	170	104	5.5	4	11	3	6
7	110	70	130	177	64	108	76.5	141	178 *176*	64	2	6.5	11	1	12.5
8	95.5	68.5	111	341.5	74.5	93.5	64	105	343	75	2	4.5	6	1.5	0.5
9	77	46.5	133	174	99.5	70.5	61	129.5	172	95	6.5	14.5	4	2	4.5
10	107.5	90	126.5	177.5	103	106.5	105.5	--- *103.0*	184 *357*	106 *83*	1	15	23.5	6.5	3
											3.3	6.1	11.6	3.9	5.95

## Data Availability

The data presented in this study are available on request from the corresponding author. The data are not publicly available due to privacy reasons.

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
