# Peer review of "Posterior Fossa Approaches Using the Leksell Vantage Frame with a Virtual Planning Approach in a Series of 10 Patients—Feasibility, Accuracy, and Pitfalls"

_brainsci, 2022, doi:10.3390/brainsci12121608_

Round 1
Reviewer 1 Report
The following is the summary of the present study:
This study explores whether these limitations can be overcome by tailored frame placement using a virtual planning approach. The posterior fossa was accessed in ten patients using the Leksell Vantage frame. Frame placement was planned with the Brainlab Elements software, including a phantom based (virtual) pre-operative planning approach. A biopsy was performed in all patients; in four, additional laser ablation surgery was performed. Accuracy of virtual frame placement was compared to actual frame placement. The posterior approach was feasible in all patients. In one case, the trajectory had to be adjusted; in another, the trajectory was switched from a right- to a left-sided approach. Both cases showed large deviations from the initially planned frame placement. A histopathological diagnosis was achieved in all patients. The new Leksell Vantage frame can be used to safely target the posterior fossa with a high diagnostic success rate and accuracy.
I think the study is original and clinically important. The outcome is promising. Furthermore, the figures are well annotated and arranged. I am happy to recommend its publication. Only some minor suggestions are list as the following:
1. The study been approved by the institutional review board. I suggest to move the information at the beginning portion of the method.
2, In Figure 3, the most left subgraphs are too small. It is very hard to know the information inside.
3. The small sample size is a limitation of the present study. Please acknowledge it.
Author Response
1. The study been approved by the institutional review board. I suggest to move the information at the beginning portion of the method.
Reply: Thank you for the suggestion. We have placed the ethical information according to the journal template. We would be happy to make the change, should the editors request that it should be moved.
2. In Figure 3, the most left subgraphs are too small. It is very hard to know the information inside.
Reply: We assume the reviewer is referring to the tables on the right. If this is the case, we do agree and thank the reviewer for pointing this out. We have enlarged the tables so that the information is easier to read. We also added the information that this is an extract of table no. 1 in the figure legend.
3. The small sample size is a limitation of the present study. Please acknowledge it.
Reply: This has been acknowledged in the Limitations section of the Discussion. However, we have added an additional sentence to further emphasise this limitation: “This does not allow for comprehensive validation of the method, but rather demonstrates its feasibility as well as the necessity to pre-plan and tailor Vantage frame-placement to each target and trajectory.”
Reviewer 2 Report
The present study about the use of a new stereotactic frame to perform posterior fossa procedure is well written. My concerns are as follow:
1) The study is about a quite small clinical series, do the authors believe that 10 cases could be enough to validate a new tool? (please explain, add details)
2) A control group is missing; actually it is not clear which frame the authors used to work with before, if also they used frameless technique and what is really the difference in between such systems in term of pros and cons of each one (please explain, add details).
Author Response
1) The study is about a quite small clinical series, do the authors believe that 10 cases could be enough to validate a new tool? (please explain, add details)
Reply: We agree that the number of patients does not allow for comprehensive validation of this method. This was not the study intention. Therefore, we have added the following sentence in the Discussion to clarify: “This does not allow for comprehensive validation of the method, but rather demonstrates its feasibility as well as the necessity to pre-plan and tailor Vantage frame-placement to each target and trajectory”.
2) A control group is missing; actually it is not clear which frame the authors used to work with before, if also they used frameless technique and what is really the difference in between such systems in term of pros and cons of each one (please explain, add details).
Reply: The intention of this study was to demonstrate that posterior fossa approaches can be safely performed with the Vantage frame, irrespective of other stereotactic methods. Compared with other systems, the challenge with the Vantage frame is the limited window and ring angle due to the side posts, transverse post, and frame holder that cannot be adjusted. The virtual planning approach to aid planning frame placement in a tailored way was therefore introduced to address these limitations, that do not apply to other systems. Moreover, since we do not present data on other stereotactic systems, it would not be appropriate to make detailed comparisons. We do hope the reviewer agrees with this approach.